# Fluorocarbosilane-Based Protective Coatings for Concrete

**DOI:** 10.3390/ma15175994

**Published:** 2022-08-30

**Authors:** Karol Szubert, Agnieszka Dutkiewicz, Marek Nowicki, Hieronim Maciejewski

**Affiliations:** 1Faculty of Chemistry, Adam Mickiewicz University, Uniwersytetu Poznańskiego 8, 61-614 Poznan, Poland; 2Poznan Science and Technology Park, Adam Mickiewicz University Foundation, Rubież 46, 61-612 Poznan, Poland; 3Centre for Advanced Technologies, Adam Mickiewicz University, Uniwersytetu Poznańskiego 10, 61-614 Poznan, Poland

**Keywords:** fluorocarbosilane, sol-gel processes, organically modified silanes, concrete, protective

## Abstract

The effectiveness of protective coatings based on 3-(2,2,3,3,4,4,5,5-octafluoropentyloxy)propyltriethoxysilane (OFTES) in protecting concrete surfaces against water was tested. For the synthesis of OFTES, 2,2,3,3,4,4,5,5-octafluoropentanol, which is a by-product in the synthesis of poly(tetrafluoroethylene), was used. The proposed silane is a cheaper alternative to the fluorinated organosilicon compounds currently used. The coatings were deposited by the sol-gel method. As a result of the creation of chemical bonds between the concrete surface and the silane, a coating was created that permanently increases the hydrophobicity of the concrete. Fluorine chains attached to silicon atoms are an effective barrier that prevents access to water and limits its impact on the concrete surface. As a result of the proposed silanization, the concrete surface obtained a hydrophobic character at contact angles of up to 126°, and the water absorption of the concrete decreased by up to 96%.

## 1. Introduction

Reinforced concrete is a strong and hard material widely used in modern construction. Moisture is the determining factor in most of the processes that damage concrete structures. As many aggressive substances enter the interior of concrete structures along with water, the absorption of surface water by concrete is an important factor determining the stability of concrete structures [1,2,3,4]. In the absence of moisture, destructive processes do not take place or are slowed down. Therefore, by limiting the possibility of moisture transport into the concrete, it is possible to increase the durability of existing concrete structures and protect the surface of new concretes. According to the PN-EN 1504-2 standard [5], one of the effective methods to protect concrete against the harmful effects of water is hydrophobic impregnation. It is a concrete treatment method that makes its surface water-repellent. As a result of the process, the pores and capillaries are not filled, but only their walls are coated. A continuous layer does not form on the concrete surface and its external appearance remains unchanged or slightly changed. Hydrophobization leads to a radical reduction in the wettability of water on the outer surface of the concrete and pores while maintaining full gas and vapor permeability.

One of the groups of compounds most commonly used in hydrophobic preparations is alkoxysilanes [6]. Organofunctional alkoxysilanes are monomeric silicon compounds that react in the presence of water or water vapor to form polysiloxanes. As a result of a sol-gel process, they form a durable polysiloxane coating covering the surface of the pores. According to the literature [7], silanes penetrate the pores of the concrete and react with its surface, making it more hydrophobic and inhibiting the penetration of water and ions dissolved in water. In sol-gel processes, silane is hydrolyzed and alkoxy groups are hydrolyzed to form silanol groups. The unstable silanol particles then lose water and generate a silicone resin as a result of the condensation reaction. In addition, some silanol groups react with hydroxyl groups in the concrete substrate through hydrogen bonds. In the final stage of the process (during drying), hydrogen bonds are converted to siloxane bonds. Silicone resin adheres well to the substrate through chemical bonds and repels surface water [8]. The alkalinity of concrete acts as a catalyst in this reaction [9].

The hydrophobic nature of the coating is due to the presence of alkyl groups [10,11,12,13]. The coating reduces water penetration but allows water vapor to be transferred outside the concrete element [14]. As shown in the literature, the use of silanes significantly reduces the amount of water that penetrates the concrete structure, which also reduces the penetration of chlorides and reduces the risk of corrosion of the reinforcement elements of the concrete [15]. It should also be emphasized that silane treatment can not only provide a hydrophobic surface but also strengthen the surface of the substrate, preventing cracking and strengthening cracks smaller than 0.2 mm [16]. Pan et al. [7,17] presented a broad discussion of the influence of silane-based impregnants on concrete.

Dai et al. [18] used a triethoxyoctylsilane solution in propanol to silanize the concrete surface. The used system reduced the water absorption by more than 90%, with a silane consumption in the amount of 115 g/m^2^. The triethoxyoctylsilane without any solvent was applied by Zhan et al. [19] to protect concrete surfaces. The water absorption was also reduced by over 90%, but with higher silane consumption (600 g/m^2^). Christodoulou et al. [15] presented the results of long-term weathering exposure on the silanized concrete surface. The samples were taken from the bridge elements protected with a silane solution 20 years ago. The studies showed that after that time, the silanization efficiency was still around 70%. 

Fluorine-containing organosilicon derivatives have recently attracted a great deal of attention because of their possible use in the production of modern materials. Fluoroalkylsilanes used in the production of surfaces resistant to oil, dirt, and water are particularly attractive [20,21,22,23,24]. The unique properties of perfluorinated silicon compounds, in particular the low surface energy due to the presence of fluoroalkyl groups, make them very attractive for use in the production of protective coatings on the surface of various materials. 

Unfortunately, limited access to substrates and complicated synthesis limit the use of fluoroalkylsilanes. In the case of the modification of building materials, various fluorinated derivatives are used mainly for the preparation of superhydrophobic protective coatings. Husni et al. [25] used 1H,1H,2H,2H-perfluorodecyltriethoxysilane to modify the silica particles in rice husk ash, which was then sprayed onto the concrete surface. A superhydrophobic coating was obtained, while water absorption was reduced by approximately 40%. On the other hand, Facio et al. [26] took a different approach: first, SiO_2_ particles were applied to the concrete surface and then functionalized with fluorinated alkoxysilane. The authors focused on the self-cleaning properties of the obtained coatings, without examining the barrier effects. 

As mentioned before, despite the excellent effects that can be obtained by surface modification with fluorofunctional organosilicon compounds, their use is not very widespread. This is mainly due to the high price and poor availability of raw materials, as well as the complex and difficult technology of their synthesis. Our experience allowed us to design and synthesize fluorinated derivatives of organofunctional silanes [27]. Therefore, a method for the synthesis of various fluorocarbofunctional silicon derivatives [28,29,30,31,32] was developed, which is an alternative to the already existing solutions and can contribute to the wider use of these derivatives in the production of hydrophobic materials. The octafluoropentanol was used in the synthesis of OFTES. The fluorinated alcohols (obtained from Grupa Azoty S.A. Tarnów, Poland) are a by-product of the synthesis of PTFE, and it is beneficial to find ways to use them. The use of fluorinated alcohols from the synthesis of PTFE gives the opportunity to reduce the costs of the synthesis of fluorinated organosilicon derivatives. Additionally, fluorinated alcohols are cheaper and more accessible than commonly used fluoroalkyl iodides. In the proposed method of synthesis, alcohols are coupled with allyl chloride in the Williamson reaction and then subjected to the process of hydrosilylation with any derivative containing an Si-H bond. This method is universal and allows us to obtain fluorofunctional silanes, polysiloxanes, and silsesquioxanes [32]. The silane was used for the production of surface coatings for wood [33] and steel [34] that protect against the adverse effects of water. In this article, we propose its use to protect a concrete surface against water penetration.

## 2. Materials and Methods

### 2.1. Materials

The chemicals were obtained from Sigma-Aldrich (Poznan, Poland) and used without any additional preparation. Concrete samples were made of Portland type 32.5 concrete (GÓRAŻDŻE CEMENT SA, Poland) according to EN 197-1 [35], gravel with a maximum grain diameter of 16 mm, and sand with a maximum grain diameter of 4 mm (Kruszgeo, Poland). Water for the preparation of the concrete was taken from the water supply, and 3-(1,1,2,2,3,3,4,4-octafluoropentyloxy)propyltriethoxysilane (OFTES) with the formula: HCF_2_(CF_2_)_3_CH_2_O(CH_2_)_3_Si(OCH_2_CH_3_)_3_ was synthesized according to the procedure described in the literature [27]. The Silicone L6 was purchased in JKK Dystrybucja (Tychy, Poland).

### 2.2. Concrete Preparation Procedure

Concrete samples were made from a mixture containing 1295 kg/m^3^ coarse aggregate (gravel), 595 kg/m^3^ fine aggregate (sand), 380 kg/m^3^ cement, and 190 kg/m^3^ water. The free water-to-cement ratio (w/c) was set at 0.5. Fresh concrete was poured and vibrated in the form of 100 × 100 × 100 mm^3^ cubes. The cubic samples were removed from the mold after 24 h of curing and then placed in an oven (20 ± 2 °C, relative humidity (RH) ≥ 90%) for 28 days.

### 2.3. Preparation of a Hydrophobic Coating

The cured concrete samples were washed with water to remove all loose parts and dried for 5 days, after which the samples were silanized. A series of alcoholic OFTES and tetraethoxysilane (TEOS) solutions were prepared. The prepared solutions were added to ethanol with 0.5 mL of concentrated hydrochloric acid (pH ≈ 2) and 5 mL of water, and then, while vigorously stirring, the appropriate amounts of silanes (OFTES and TEOS) were added to the solutions. The amount of ethanol was chosen to obtain 500 mL of solution each time. The amounts of silanes added to the solutions are shown in Table 1.

The solutions were mixed for 1 h, after which time the concrete samples were completely immersed in the solutions for 1 h. After removal from the solution, the samples were dried at room temperature for 24 h.

The consumption of the silane solution was calculated from the weight loss of the OFTES solution (g) by the silanized surface area (m^2^).

In order to evaluate the effect of hydrochloric acid on the concrete surface, an additional F5* solution was prepared. After one hour of stirring, the solution was neutralized (to pH = 7) with an ammonium hydroxide solution. A concrete sample was placed in the solution immediately after being neutralized. The previous procedure was then followed.

### 2.4. Analyses and Measurements

Scanning electron microscopy (SEM) (FEI Company, Hillsboro, OR, USA) images were taken on an FEI Quanta 250 FEG microscope equipped with an EDAX Energy Dispersive Spectroscopy detector (EDS) (EDAX, Pleasanton, CA, USA). The images were taken in high vacuum mode with 10 kV accelerating voltage. EDS mapping was performed with an electron beam energy of 20 keV using the EDS Octane SDD detector (EDAX). The SEM images were recorded for concrete samples cut to the size of a cube of 10 × 10 × 10 mm and subjected to silanization. The concrete samples were prepared for SEM imaging by gluing them onto standard SEM carbon adhesive tape. 

Static water contact angle (WCA) measurements (A.KRÜSS Optronic GmbH, Hamburg, Germany) on all samples were made using a Krüss GmbH DSA 100 Expert Drop Shape Analyzer equipped with a software-controlled (DAS4 2.0): x, y, z-axis table, a quadruple-dosing unit with zoom and focus adjustment, illumination, and a camera with 780 × 580 px resolution. All the data presented are arithmetic means of measurements made for 5 drops per sample. The measurements of contact angles were performed immediately after the deposition of a drop on a studied surface. In an additional experiment on the surface of the concrete sample, a drop of water containing a dye was placed to observe the hydrophobic character of the surface; the drop containing a 0.1% solution of methyl orange had a volume of 20 µL.

To determine the water absorption of concrete cubes, a liquid water permeability test was used. The test was carried out according to the EN 1062-3 standard [36]. Before the actual tests, the concrete samples were conditioned in three cycles of drying and immersion for 24 h each. The samples were weighed, to the nearest 0.1 g, before immersion in water. The water level was 5–10 mm above the top of the concrete sample. The test sample was supported on a plastic rack with 10 mm space above the base of the container. All air bubbles from the concrete surface were removed by careful wiping with a clean, damp, lint-free cloth 10 min after the start of the test. After 24 h, the test specimen was removed from the water, carefully wiped dry using absorbent paper, and weighed to the nearest 0.1 g. The relative permeability of the liquid water for 24 h (w) was calculated using Equation (1):(1)$$w=\frac{∆m_{}}{\surd 24\times A}\;\left[\mathrm{kg}/(m^{2}\times h^{0.5}\right)]$$
where Δ*m* is the mass variation before and after submersion (kg); *A* is the surface area of the sample (m^2^).

The penetration depth of the silanization solution was determined according to the EN-1504-2 standard [5]. To determine the depth of the penetration of the silanization solution in the concrete sample, after silanization, the concrete cubes were fractured into two parts. Then, the fracture surface was sprayed with water; the border between the bright area (the silanized region not wetted by the water) and the dark area (the region wetted by the water) marks the depth of penetration of the silanization solution into the concrete.

## 3. Results and Discussion

Fluorocarbosilane (OFTES) has been used to create protective coatings that reduce the penetration of water into concrete. These coatings were made on the basis of the sol-gel process. As a result of the hydrolysis and condensation reactions taking place in the system, chemically bonded to the hydroxyl groups present on the concrete surface, a siloxane coating was obtained. According to current knowledge [7], silanes do not form coatings on the concrete surface, but penetrate into concrete pores and react with their surface; additionally, the alkaline nature of concrete has a positive effect on the sol-gel process [8]. The way of creating protective coatings based on OFTES is presented in Figure 1. In the first stage, the alkoxysilane is hydrolyzed; the acidification of the system (through the addition of hydrochloric acid) accelerates this process. The pH of the acidified system is about 2. Based on the ^29^Si MNR analysis, it was found that after 1 h, total hydrolyzed silane and small amounts of partially condensed siloxanes are present in the system [37]. In the next step, the obtained silanol reacts with other silanol molecules and with the hydroxyl groups on the concrete surface. After the concrete samples are removed from the solution, the solvent evaporates quickly, which has a positive effect on the condensation of the system. Additionally, to determine the effect of the hydrochloric acid on the concrete surface, an additional test was carried out: after the hydrolysis step, the solution was neutralized (with ammonium hydroxide), and a concrete sample was immersed into a neutral sol. The effect of neutralization on the protective coating will be discussed later in this work.

The protective coatings were obtained by immersing concrete samples in silanizing solutions. The solutions used can be divided into two groups: solutions containing only one silane (OFTES) and solutions prepared with the addition of tetraethoxysilane (TEOS). A series of solutions containing 1 to 5% of the discussed silanes was prepared. The prepared solutions were additionally acidified with hydrochloric acid to accelerate the hydrolysis reaction. Before the concrete samples were immersed, the solutions were mixed for 1 h. This time was selected on the basis of previous studies of sol-gel processes that take place in the presence of OFTES [33,34]. The average consumption of the OPTES in the immersion method was 430 ± 27 g/m^2^, calculated for the OPTES solutions.

Figure 2 shows SEM images of the concrete surface before and after silanization with solutions containing various amounts of OFTES. Pure, unmodified concrete creates complex, porous structures with a relatively uniform surface. This is especially visible in the image taken at lower magnification (Figure 2a). As mentioned above, the modification of the concrete surface with silane does not block its porous structure. Images taken of concrete samples after silanization show additional layers that cover the concrete surface, especially visible at higher magnifications. Only in the case of silanization with the F5 solution obtained from the OFTES solution with the highest concentration (Figure 2h), a continuous protective coating was obtained. In the case of the other samples (F1 and F2.5), the protective coatings produced did not form a continuous protective layer. 

These observations are also confirmed by the EDS mapping images of silicon and fluorine atoms presented in Figure 3. The main element on the surface of unmodified concrete is calcium, while in the case of samples after silanization, silicon dominates. The fluorine atom mapping images confirm the homogeneous distribution of fluorine in the protective coatings created on the concrete surface. Although the distribution of fluoride atoms is even, it does not clearly confirm that fluorocarbosilane creates a continuous layer. Additionally, crystallites with a regular structure were observed on the concrete surface, with a particularly large amount observed on the concrete surface in sample F1. On the basis of the analysis of the EDS mapping images (see additional materials), these are inorganic chlorides (probably KCl) that crystallized on the concrete surface (exemplary EDS mapping images of selected elements are also included in the Appendix A). Crystallites are formed as a result of the dissolution/partial etching of the concrete surface by hydrochloric acid from silanizing solutions. It should be noted that similar crystallites on the concrete surface were not observed in the case of silanization with the use of octyltriethoxy resin [37]. On the other hand, in the case of silanization with organosilicon derivatives of fatty acids, shapeless silicate structures were formed on the concrete surface (identification based on EDS mapping) [9].

Figure 4 shows SEM images of the surface of concrete subjected to silanization in solutions containing the TEOS additive. When solutions containing the TEOS additive are used, the obtained coatings create highly developed, porous structures, and the concrete surface is visible in the “holes”. The use of higher concentrations of OFTES improves the continuity of the protective coatings; on the other hand, the introduction of TEOS into silanizing solutions favors the production of coatings with greater porosity. In this case, the SEM images of the concrete surface also show crystallites, most often with regular shapes. 

The EDS mapping images of the silicon and fluorine atoms also confirm the dominant role of these elements on the surface of concrete subjected to silanization with solutions containing the TEOS additive. The surface of sample F5_5 silanized in a solution containing 5% OFTES and 5% TEOS is characterized by the highest content of fluorine (Figure 5b). However, the distribution of fluorine atoms in the coatings is uneven, with a large share of dark areas. These areas overlap with the “holes and cracks” observed in the SEM images. Based on the SEM and EDS analyses, it can be concluded that TEOS had a negative impact on the coatings. 

Table 2 shows the results of the contact angle measurements made for the silanized concrete samples and for the unmodified concrete samples. As a result of the strongly hydrophilic nature and high water absorption, it was impossible to measure the contact angle on the surface of the unmodified concrete; the drop of water was immediately absorbed by the concrete (see Figure 6). Moreover, in the case of sample F1_4, it was not possible to carry out the measurement—the drop immediately spread over the surface of the concrete. In solution F1_4, TEOS dominated as a silica precursor and the amount of the fluorinated derivative was not enough to obtain the hydrophobic character of the surface. In the case of the other samples, the obtained results of the measurement of the contact angle confirm the hydrophobic character of the modified surfaces, and contact angles above 116° were obtained. Taking into account the standard deviations of the individual measurements, it can be concluded that all samples (except F1_4) were of a similar hydrophobic nature. Furthermore, it should be emphasized that the measurements were made immediately after the drop was applied to the analyzed surface, in accordance with the procedure for static contact angle measurements. Figure 6 shows the images recorded during the WCA measurements. As mentioned earlier, a drop of water placed on the surface of the raw concrete was immediately absorbed and the measurement was impossible to perform (see Figure 6a). Figure 6e shows an image of a single measurement taken for sample F5. To better visualize the behavior of the droplet on the silanized surface, images were taken. The images show how a drop of water on the concrete surface behaves over time. The images were taken immediately after the drops were applied, after some time, and after the water had completely evaporated. In the case of unmodified concrete, the water immediately spread over the surface of the concrete and was absorbed very quickly. After 2 h, it was completely dry, and an irregular stain was visible (left by methyl orange). In the case of the silanized concrete surface, the drop of water remained on the concrete surface the whole time. Even after 5 min, the drop shape did not change (did not spread over the concrete surface and was not absorbed by the concrete). After the water had completely evaporated, a regular stain of dye remained on the surface. 

Table 3 presents the results of the water absorption measurements made according to the EN 1062-3 standard [36]. The results obtained for unmodified concrete are similar to the data in the literature [38]. However, the silanization process reduced water absorption by at least 81.1% (the worst result, for sample F1_4). Effective protection against water penetration was guaranteed by samples modified with solutions containing 2.5% OFTES (reduction in water absorption by at least 92.49%). A further, twofold increase in fluorinated silane concentration (increase in silane consumption from 11.40 g/m^2^ to 20.42 g/m^2^) reduced water absorption by almost 96% compared to unmodified concrete. As expected, it was observed that the barrier properties increased with increasing OFTES concentration. The addition of TEOS does not improve the barrier properties of the coatings. In systems containing identical amounts of OFTES, higher water absorption was obtained for systems containing TEOS. It is more advantageous to use systems that contain only OFTES.

The penetration depth of silane (silanizing solution) into the concrete is presented in Table 4. As mentioned earlier, the measurements were made in accordance with the EN-15-2 standard [5]. As intended, the surface covered with a layer of silane does not absorb water and is lighter in color, while the deeper layers of concrete perfectly penetrate the water and are darker. Unfortunately, in most cases, for concrete samples silanized with OFTES solutions, the boundaries between the lighter and darker regions are not clear. The best visible boundary between the regions was observed for sample F5; see Figure 7. The penetration depth was around 3–10 mm for all samples. Only in the case of sample F1_4, the entire concrete surface was uniformly wetted by the water. Similar observations were described earlier in the section on measuring WCA. In the case of concrete samples modified in solutions containing only OFTES, it can be seen that when the silane concentration increases, the depth of penetration into the concrete structure also increases. The penetration depth for samples F1, F2.5, and F5 increased from 3–5 mm to 6–7 mm, respectively. Moreover, the addition of TEOS increased the penetration depth of the solutions into the concrete. In the case of sample F5_5, the solution penetrated the concrete structure at 9–10 mm.

As mentioned above, the formation of KCl crystals on the surface of silanized concrete is due to the presence of relatively large amounts of HCl in the solution. Concrete silanization with a neutral 5% OFTES solution was also carried out. After 1 h of stirring, the acidic solution was neutralized with ammonium hydroxide solution to pH ≈ 7. A concrete sample was immediately introduced into the neutralized solution, followed by the original procedure. The SEM and EDS images of the sample containing 5% OFTES after neutralization (F5*) are shown in Figure 8. In the SEM images at high magnification, cracks in the continuous layer covering the concrete surface are clearly visible. The EDS image mapping of silicon atoms shows brighter areas corresponding to different concentrations of organosilicon compounds. Probably after neutralization, the condensation process begins to dominate, and the concrete surface is affected by compounds with longer, more complex siloxane chains, which has a negative effect on the produced coating. 

Table 5 shows the results of the WCA measurements and water absorption properties of the F5* sample. The results of the WCA measurements are similar to the results obtained for F5, the neutralization of the system did not have a significant effect on the hydrophobic nature of the silanized concrete surface. The water absorption of the F5* sample is approximately two-fold greater than the F5 sample, the barrier effect of the created coating is weakened, and due to the cracks visible in the SEM images, water can penetrate into the internal concrete structures faster. 

The stability of the obtained coatings was tested on the basis of immersing and drying concrete samples. Five cycles of immersing and drying were performed. The samples were first immersed in water for 24 h and then dried for 5 days. In addition, studies were also carried out with the commercially available Silicone L6 in order to compare the results. The results are summarized in Table 6. The water absorbability of raw concrete was relatively high and constant in all the immersion cycles, and the absorbed water was easily evaporated during the drying stage. As a control, Silicone L6 was used, which was applied twice with a brush to the concrete surface according to the manufacturer’s instructions. The Silicone L6-protected sample showed relatively high water absorption (water permeability 0.0964) in the first immersion cycle. In subsequent cycles, the water permeability was more than two times lower, at a level of 0.04. In addition, in all drying cycles, water evaporates almost completely. In the case of concrete protected with a 5% OFTES solution (F5), in all the immersion cycles, the results were similar to those presented earlier (see Table 3). The protective coatings based on OFTES are stable and the barrier properties did not deteriorate in the subsequent cycles. Furthermore, in the drying cycle, all the absorbed water evaporated, the coating was permeable to water vapor, and did not restrict “the breathing of the concrete”.

## 4. Conclusions

The effectiveness of OFTES-based protective coatings in protecting concrete surfaces against water was tested. As a result of the sol-gel process, chemical bonds between the concrete surface and the silane are created; the obtained coatings increase the hydrophobicity of concrete. As a result of the silanization, the water absorption of concrete decreased by up to 96%. Only a small amount of OFTES (2.5 and 5%) reduced the water absorption of concrete by more than 90%. It was also shown that OFTES creates stable coatings and the water absorption did not change in the subsequent cycles of the immersion and drying of concrete samples. It has been shown that the neutralization of silanizing solutions has a negative effect on the produced coatings. 

## Figures and Tables

**Figure 1 materials-15-05994-f001:**
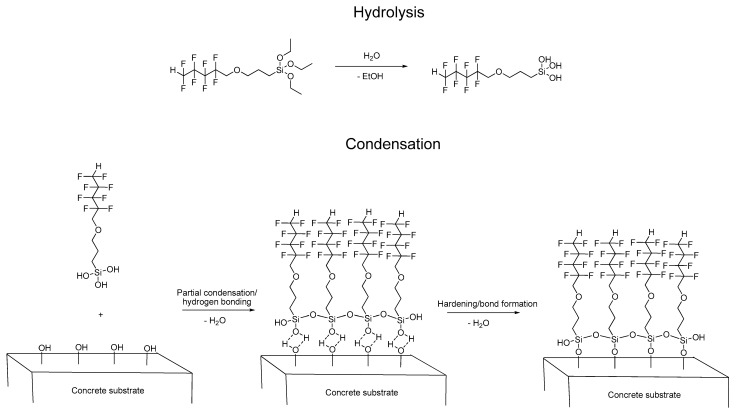
The sol-gel reaction mechanism between OFTES and the concrete surface.

**Figure 2 materials-15-05994-f002:**
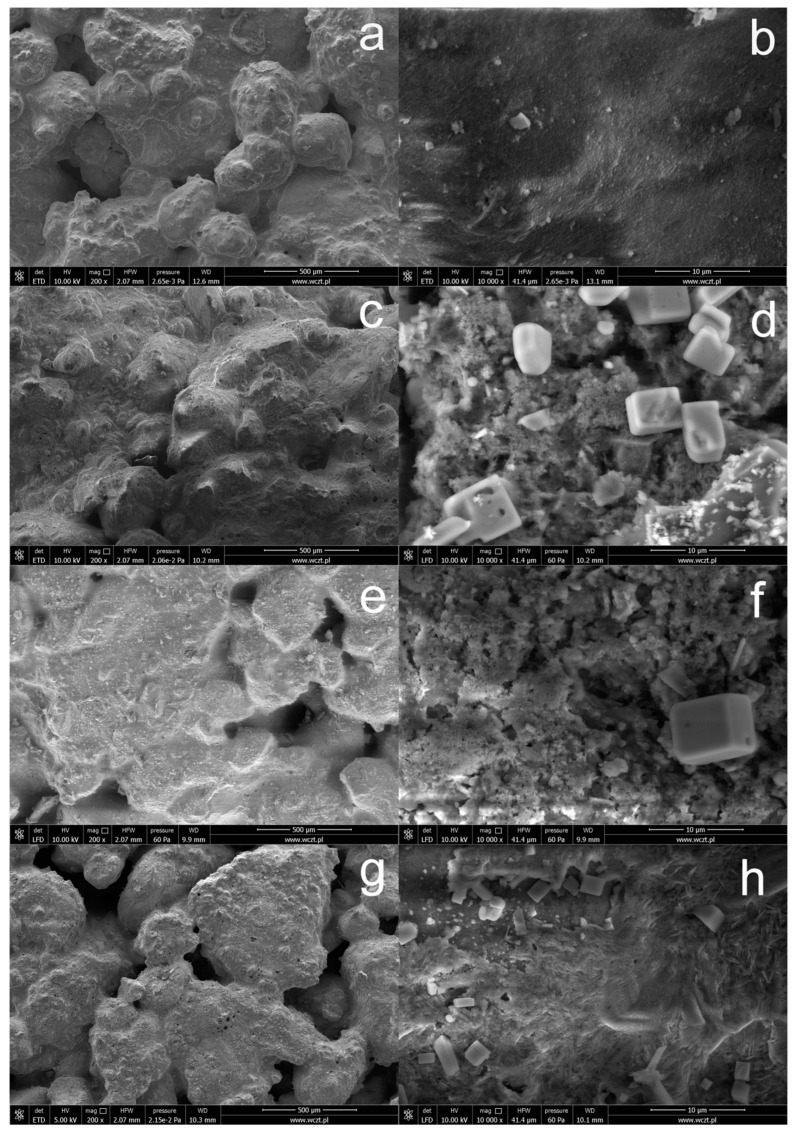
SEM images of unmodified concrete (**a**,**b**) and concrete with a modified surface by OFTES solutions: sample F1 (**c**,**d**), F2.5 (**e**,**f**), and F5 (**g**,**h**).

**Figure 3 materials-15-05994-f003:**
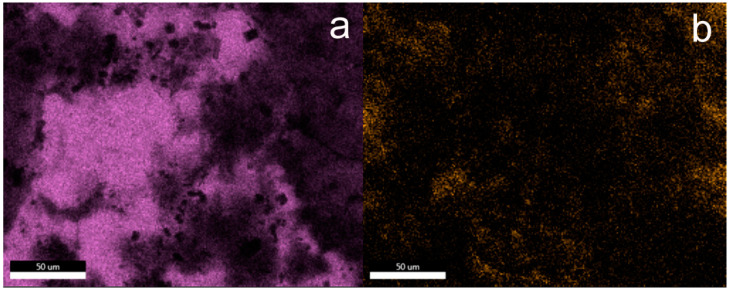
EDS mapping of the elements on the F5 surface; element map of Si (**a**) and F (**b**).

**Figure 4 materials-15-05994-f004:**
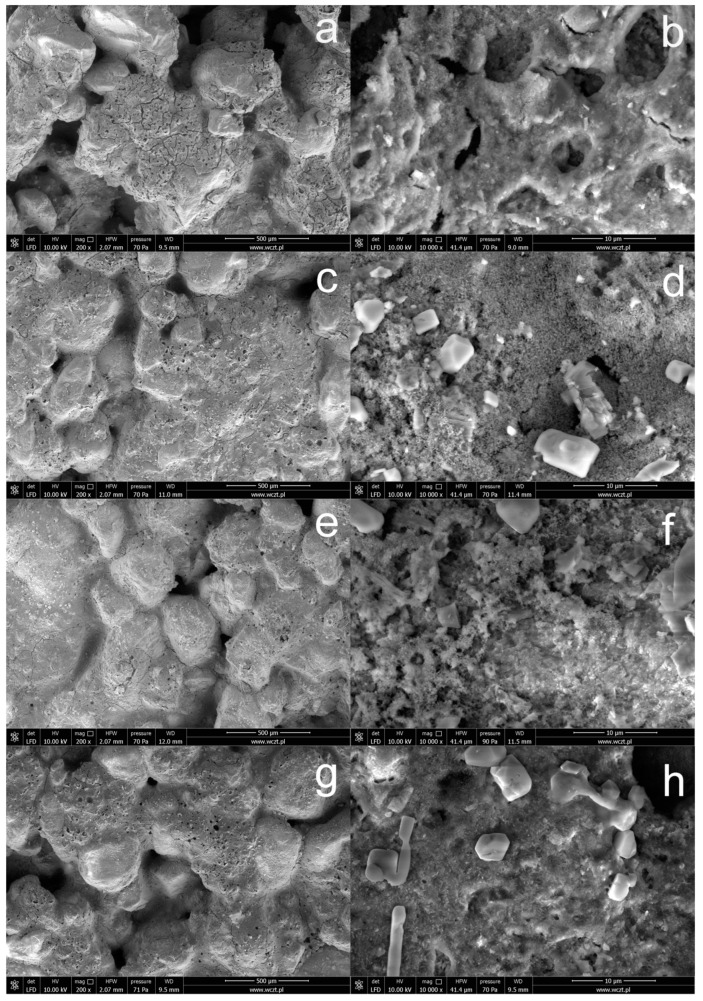
SEM images of concrete with a modified surface by OFTES/TEOS solutions: sample F1_4 (**a**,**b**), F2.5_2.5 (**c**,**d**), F4_1(**e**,**f**), and F5_5 (**g**,**h**).

**Figure 5 materials-15-05994-f005:**
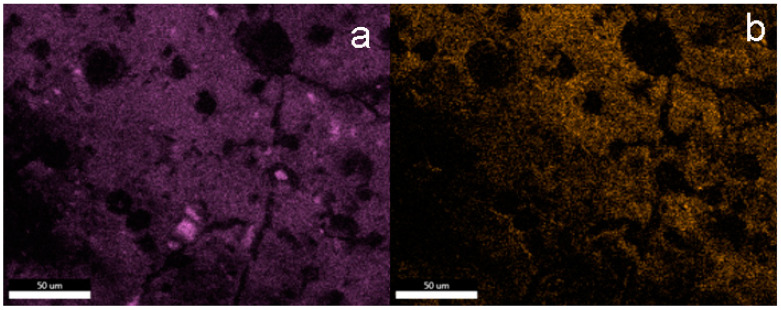
EDS mapping of the elements on the F5_5 surface; element map of Si (**a**) and F (**b**).

**Figure 6 materials-15-05994-f006:**
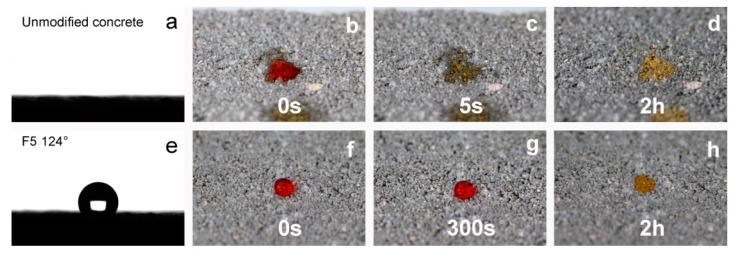
The WCA on the unmodified concrete surface (**a**) and with coating F5 (**e**); images of a water droplet on the unmodified surface (**b**–**d**) and with coating F5 (**f**–**h**); directly after deposition and after evaporation.

**Figure 7 materials-15-05994-f007:**
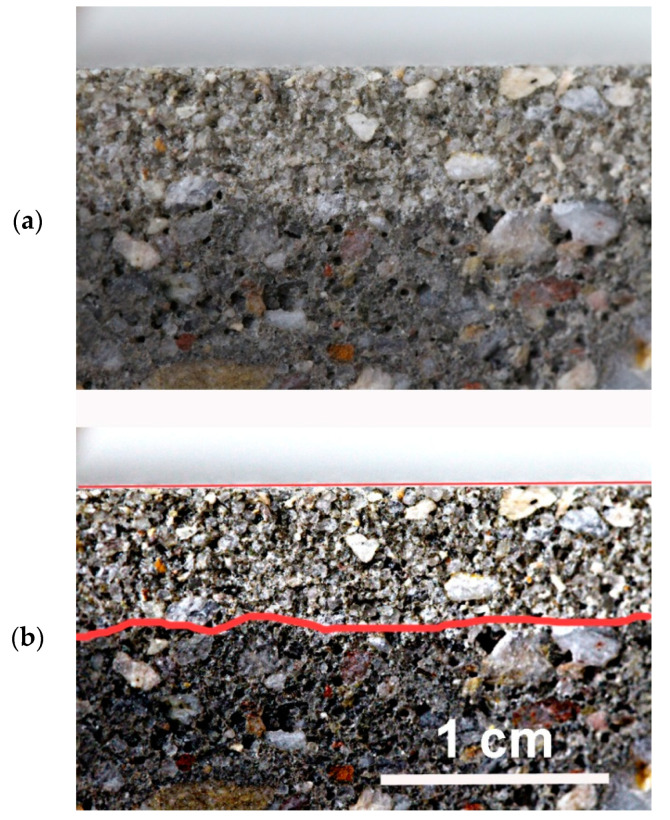
Results of penetration depth revealed by water spraying: raw photo (**a**) and photo after contrast adjusting (**b**).

**Figure 8 materials-15-05994-f008:**
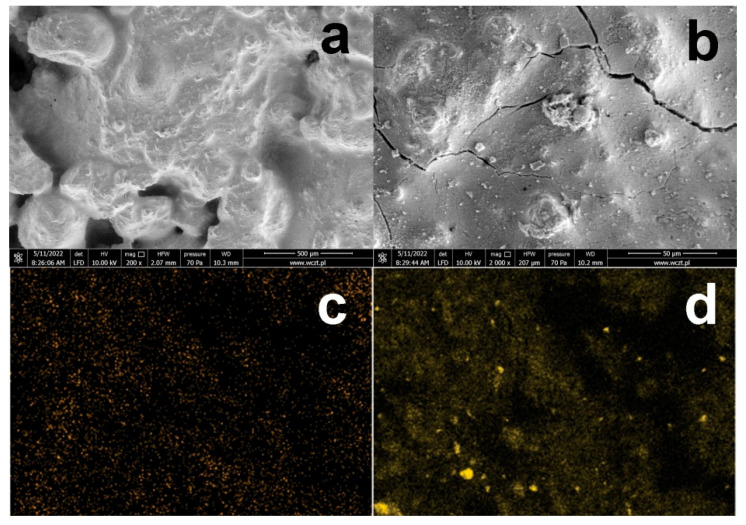
SEM images of concrete with a modified surface by neutral 5% OFTES solutions (**a**,**b**); EDS mapping of the elements on the F5* surface; element map of F (**c**) and Si (**d**).

**Table 1 materials-15-05994-t001:** Amounts of reagents used for the preparation of silane solutions.

Sample Name	OFTES [mL]	TEOS [mL]
F1	5	0
F2.5	12.5	0
F5	25	0
F1_4	5	20
F2.5_2.5	12.5	12.5
F4_1	20	5
F5_5	25	25

**Table 2 materials-15-05994-t002:** Water contact angle (WCA) values of concrete without and with OPTES treatment.

Sample Name	WCA [°]
Unmodified concrete	-
F1	116.4 ± 3.7
F2.5	126.2 ± 2.8
F5	122.9 ± 3.1
F1_4	-
F2.5_2.5	125.5 ± 7.6
F4_1	124.1 ± 4.1
F5_5	118.1 ± 5.9

**Table 3 materials-15-05994-t003:** Relative water permeability (w) after 24 h of the immersion of concrete samples without and with silane treatment.

Sample Name	w [kg/m^2^ h^0.5^]	Relative Improvement in Absorbability [%]	Consumption of the OPTES (Consumption of the OPTES Solution) [g/m^2^]
concrete	0.4777 ± 0.0420	-	-
F1	0.0853 ± 0.0043	82.15	4.21 (420.54)
F2.5	0.0303 ± 0.0020	93.66	11.40 (455.85)
F5	0.0196 ± 0.0012	95.90	20.42 (408.46)
F1_4	0.0903 ± 0.0071	81.10	4.08 (408.46)
F2.5_2.5	0.0359 ± 0.0027	92.49	11.43 (457.08)
F4_1	0.0323 ± 0.0024	93.24	15.98 (399.38)
F5_5	0.0255 ± 0.0026	94.65	23.12 (462.31)

**Table 4 materials-15-05994-t004:** The penetration depth of the silane into the concrete.

Sample Name	Penetration Depth [mm]
concrete	-
F1	3–5
F2.5	5
F5	6–7
F1_4	-
F2.5_2.5	5–6
F4_1	5–6
F5_5	9–10

**Table 5 materials-15-05994-t005:** The concrete with neutral 5% OPTES treatment: water contact angle (WCA) values and relative water permeability (w) after 24 h of immersion.

Sample Name	WCA [°]	w [kg/m^2^ h^0.5^] (Relative Absorbability [%])
F5*	118.4 ± 2.3	0.0385 ± 0.0032 (91.94)

**Table 6 materials-15-05994-t006:** Relative water permeability (w) after 24 h of the immersion of concrete samples without and with silane treatment.

Sample Name	w [kg/m^2^ h^0.5^]				
	1	2	3	4	5
Concrete	0.4789	0.4669	0.4608	0.4505	0.4556
Silicone L6	0.0964	0.0465	0.0462	0.0421	0.0437
F5	0.0202	0.0197	0.0182	0.0191	0.0189

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
