# Peer review of "Fluorocarbosilane-Based Protective Coatings for Concrete"

_materials, 2022, doi:10.3390/ma15175994_

Round 1
Reviewer 1 Report (Previous Reviewer 3)
The authors present a work concerning the preparation of hydrophobic coatings for concrete. This article is very well documented, clear and with convincing results. Therefore, I propose to accept the article as is. I only ask to replace "table below" by "table 1" (page 3) and "scheme 1" by "figure 1" (page 5).
Author Response
Response to reviewers
We would like to kindly thank our Editor and Reviewers for their valuable remarks and critics.. We hope our manuscript is close to being finalised.
Reviewer #1
The authors present a work concerning the preparation of hydrophobic coatings for concrete. This article is very well documented, clear and with convincing results. Therefore, I propose to accept the article as is. I only ask to replace "table below" by "table 1" (page 3) and "scheme 1" by "figure 1" (page 5).
Comment
I only ask to replace "table below" by "table 1" (page 3) and "scheme 1" by "figure 1" (page 5).
Reply
The manuscript have been corrected according to the Reviewer suggestion.
Reviewer 2 Report (New Reviewer)
The paper adopts the traditional method of silane impregnation to improve the hydrophobicity of concrete materials so as to protect concrete. As a kind of coating, it should be easy to apply, if only using impregnation method, it is difficult to apply in practical concrete engineering, the author should explore the spraying or brush coating process of protective coating.The paper also needs to supplement from the following aspects :(1) the amount of organosilane per unit area;(2) the depth at which silane diffuses into concrete;(3) mechanical strength of silane protective layer, such as friction resistance.
Author Response
Response to reviewers
We would like to kindly thank our Editor and Reviewers for their valuable remarks and critics. Additionally, the entire manuscript has been re-checked for English.
We hope our manuscript is close to being finalised.
Reviewer #2
The paper adopts the traditional method of silane impregnation to improve the hydrophobicity of concrete materials so as to protect concrete. As a kind of coating, it should be easy to apply, if only using impregnation method, it is difficult to apply in practical concrete engineering, the author should explore the spraying or brush coating process of protective coating.The paper also needs to supplement from the following aspects: (1) the amount of organosilane per unit area;(2) the depth at which silane diffuses into concrete;(3) mechanical strength of silane protective layer, such as friction resistance.
Comment
(1) the amount of organosilane per unit area
Reply
Information on the amount of silane has been added.
Comment
(2) the depth at which silane diffuses into concrete
Reply
Information on the depth of silane penetration into concrete is provided in the manuscript; Table 4 and Figure 7.
The penetration depth of the silanization solution was determined according to the EN-1504-2 standard. The concrete cubes were fractured in two parts. Then the fracture surface was sprayed with water, the surface covered with a layer of silane does not absorb water and is lighter in color, while the deeper layers of concrete perfectly penetrate the water are darker.
Comment
(3) mechanical strength of silane protective layer, such as friction resistance
Reply
Mechanical strength tests have not been the subject of our research. At the moment, I do not have the ability to quickly perform this type of test. Thank you very much for your suggestion, I will definitely add strength tests in further coating tests.
Reviewer 3 Report (New Reviewer)
Manuscript ID: materials-1842538
Title: Fluorocarbosilane-based protective coatings for concrete
The following amendments are recommended:
1. The novelty of the research and its difference from previous works should be written at the end of the introduction section.
2. Images of experimental work should be provided.
3. EN 1062-3 mentioned that the concrete samples are subjected to three cycles of drying and immersion for 24 hours each, which was not mentioned in section 2.4. Please clarify.
4. You should add more elaborations to the SEM images in Figures 2 and 4.
5. Most hydrophobic materials lose a part (or sometimes all) of their properties over time. As is known, the service life of concrete is relatively long, which may exceed tens of years. The question about the feasibility of using such materials to coat concrete, according to the above.
6. The conclusions section contains a lot of general information. It should be shortened and focus on the main results.
Author Response
Response to reviewers
We would like to kindly thank our Editor and Reviewers for their valuable remarks and critics.. We hope our manuscript is close to being finalised.
Reviewer #3
Manuscript ID: materials-1842538
Title: Fluorocarbosilane-based protective coatings for concrete
The following amendments are recommended:
Comment
The novelty of the research and its difference from previous works should be written at the end of the introduction section.
Reply
Information on the advantages of using OFTES has been moved to the end of the introduction section.
Comment
Images of experimental work should be provided
Reply
Unfortunately, we did not keep photographic documentation during the tests, only the experiments on the behavior of water drops on the surface of the concrete and the determination of the penetration depth of the concrete contain some photos; these photos were already included in the manuscript.
Comment
EN 1062-3 mentioned that the concrete samples are subjected to three cycles of drying and immersion for 24 hours each, which was not mentioned in section 2.4. Please clarify.
Reply
Thank you very much for this comment; the description of the samples conditioning according to EN 1062-3 has been added.
Comment
You should add more elaborations to the SEM images in Figures 2 and 4.
Reply
Comment
Most hydrophobic materials lose a part (or sometimes all) of their properties over time. As is known, the service life of concrete is relatively long, which may exceed tens of years. The question about the feasibility of using such materials to coat concrete, according to the above.
Reply
The service life of a protective coating is a very important issue. However, conducting long-term tests was not the subject of this manuscript. The stability of the obtained coatings on the basis of immersing and drying concrete samples were carried out. See Table 7. In the Introduction section we mentioned the results of long-term weathering exposition on the silanized concrete surface presented by Christodoulou et al. [15]. The studies showed that after 20 years the silanization efficiency was still around 70%.
Comment
The conclusions section contains a lot of general information. It should be shortened and focus on the main results.
Reply
The conclusion section has been corrected.
Round 2
Reviewer 3 Report (New Reviewer)
The authors have tried to improve their manuscript. No further modifications are needed.
Author Response
Dear Reviewer,
Thanks a lot for your earlier comments. I am glad that the manuscript in its current form is satisfactory.
This manuscript is a resubmission of an earlier submission. The following is a list of the peer review reports and author responses from that submission.
Round 1
Reviewer 1 Report
In this paper, the fluorocarbosilane-based protective coatings for concrete were designed with the aim of reducing moisture damage. However, there are still some issues that should be further explained. Thus, I think a major revision is needed, and the specific comments are as follows.
- In Abstract and Introduction: Referring to available literature, the lack of using fluorocarbosilane in protecting concrete cannot be applied as the necessity of this research. The authors need to redesign the abstract and introduction with the aim of obtaining the fundamental innovation of this paper.
- Section 2.1: The detailed material properties of propyltriethoxysilane should be added, and what does the chemical structure be like? Is there any chemical reaction between the propyltriethoxysilane and concrete material?
- Section 2.3: Lack of real picture related to the preparation of a hydrophobic coating.
- How about the long-term durability of the hydrophobic coating. Authors need to conduct this analysis, which is vital to evaluate the performance of the hydrophobic coating.
- What is the cost of the hydrophobic coating?
- Is there any comparison of the hydrophobic effects with other research?
- What is the mechanism of the hydrophobic coating in against the water?
- In summary, this paper is too simple, and the authors need to add more contents to improve the quality of this paper.
Reviewer 2 Report
Very poor quality of work, apart the lack of originality. Results are not useful and are also unreliable. All formulations approximately the same results as the control. No real characterization of systems was carried out.
Here are some specific problems:
- Author specify that local water was used, but gave no information about pH, conc. of minerals etc. They state that conc HCl was used but did not check the pH of the solution. This is crucial for sol-gel reactions.
- Contact angle was measured on highly substrates that causes water to drain rapidly
- Values for penetration depth are the opposite to what it would have been expected because the lower surface energy of systems containing the fluorinated silanized coating should give a lower capillary drive for the water penetration.
- Authors do not attempt to explain the results.
Reviewer 3 Report
Authors describe in this article the preparation of hydrophobic coatings for concrete using a sol-gel process from organosilanes as reactants. The introduction is well documented, clear and complete.
The results are generally well presented, however I recommend that the authors submit a new revised version on the following points.
1) After consulting the supplementary material and the text on pages 5 to 8, it seems difficult to make a connection, as the authors say, between continuity of protective layers and microscopy images/EDS analysis. For instance for F2.5 and F5, it is not obvious, according to EDS of F and Si to assert that the layer for F5 is continuous and that for F2.5 is not continuous. This discussion should be improved by properly paralleling the EDS analyses with the microscopic observations. To include some EDS images (for fluorine and silicon) in the text seems to me essential for a good understanding.
2) Page 11: The method for determining the penetration depth of silane is simply given by a reference, i.e. “standard method” (EN-15-2) [5]. It would be of interest to quickly present the method and possibly show an image of the light and dark areas showing the areas of the concrete covered or not by the hydrophobic coating.
3) In addition, the conclusion needs to be completely revised. Important results are not well highlighted. In my opinion, it should also contain perspectives to the work presented by the authors.
4) Finally, all ml must be replaced by mL. A bracket is missing after h^0.5 in the units of the formula (1).